# DocCPLNet: Document Image Rectification via Control Point and Illumination Correction

**DOI:** 10.3390/s25206304

**Published:** 2025-10-11

**Authors:** Hongyin Ni, Jiayu Han, Chiyuan Wang, Shuo Zhang, Ruiqi Li

**Affiliations:** School of Computer Science, Northeast Electric Power University, Jilin 132012, China; nihongyin@neepu.edu.cn (H.N.); 2202400878@neepu.edu.cn (C.W.); 2202400871@neepu.edu.cn (S.Z.); 2202400846@neepu.edu.cn (R.L.)

**Keywords:** geometric distortions, control points, spatial attention, geometric unwarping, illumination correction

## Abstract

With the widespread adoption of mobile devices in daily life, efficiently capturing and digitizing documentation has emerged as a critical research question. The acquisition of documents via mobile devices is often compromised by shadow interference and geometric distortions, which degrade image quality and adversely affect both OCR accuracy and readability. To address this, we propose a novel method that utilizes control points and illumination prediction to effectively rectify distortions and eliminate shadows in captured document images. Spatial attention is employed to guide the interpolation between control points and reference points, effectively eliminating geometric distortions in the captured document images. Following geometric unwarping, an illumination correction model is applied to remove shadows and enhance surface clarity, improving both human readability and OCR accuracy. Our method demonstrates robust performance in effectively rectifying document distortions across diverse scenarios. Evaluation on the DocUNet benchmark dataset shows that our approach achieves competitive results compared with state-of-the-art techniques.

## 1. Introduction

The rapid advancement of digital transformation has led to the widespread adoption of mobile devices such as smartphones as ubiquitous tools for capturing and digitizing paper-based documents. This shift enables convenient on-the-go digitization, facilitating the transition from physical to electronic information management. However, during the document acquisition process, perspective and geometric distortions often arise due to suboptimal camera angles as well as inherent physical deformations of the paper such as folds, curls, or wrinkles. Additionally, uneven illumination, shadows, and surface contaminants further degrade image quality, impairing textual clarity and structural integrity. These issues pose significant challenges for downstream applications, including information retrieval, content analysis, archival preservation, and document editing, ultimately hindering efficient digitization workflows. To mitigate the impact of such distortions, numerous approaches have been proposed in prior research with the aim of enhancing the quality and usability of captured document images.

Early studies on document geometric rectification have primarily relied on auxiliary hardware [1,2,3] to correct captured document images. However, such hardware-dependent approaches introduce additional equipment requirements, limiting their portability and practicality in casual settings. With the advent of deep learning, Ma et al. [4] pioneered the first method using stacked U-Nets to predict pixel-wise displacements, effectively addressing document surface creases and curvatures. While this approach demonstrated promising results, its performance degrades under complex backgrounds due to insufficient spatial context modeling. To enhance flexibility, Xie et al. [5] proposed a Thin-Plate Spline (TPS [6]) interpolation framework that regresses sparse control points to reference points, enabling adaptive rectification with adjustable mesh resolution. Although efficient, this method struggles with highly nonlinear distortions. Later, Das et al. [7] introduced a 3D-to-2D regression pipeline that explicitly models document geometry through 3D coordinate prediction, significantly improving physical plausibility but requiring computationally expensive synthetic training data. Most recently, Xue et al. [8] presented a Fourier-space transformation approach that separates high-frequency textual content from low-frequency background noise, achieving robust restoration under photometric degradation. Despite its innovation, this method faces challenges when document backgrounds exhibit text-like patterns, leading to false rectification. Early illumination correction methods predominantly relied on intrinsic image concepts such as light contrast [9] and albedo distribution [10]. Recent advances in deep learning have introduced innovative solutions to address these challenges. For instance, Li et al. [11] pioneered a patch-based CNN framework that decomposes distorted document images into six of overlapping patches, predicts local geometric flows, and stitches results via gradient-domain optimization to minimize artifacts. While this approach efficiently handles moderate folds and curvature, its reliance on synthetic training data limits generalization to real-world distortions, and patch-wise processing occasionally introduces stitching errors in textureless regions. Building on this, Feng et al. [12] proposed DocTr, a transformer-based architecture leveraging self-attention mechanisms to model global document deformation and illumination variations. This method achieves superior illumination correct but suffers from high computational costs and sensitivity to extreme color distortions. In contrast, Wang et al. [13] developed UDoc-GAN, an unpaired adversarial framework that integrates ambient light priors to guide illumination correction without requiring paired training data. Although it preserves textual details and enables real-time processing, its performance hinges on ambient light diversity in training and struggles with crease-induced localized shadows. This work presents a comprehensive solution for document distortion correction by integrating geometric rectification with illumination refinement. Our approach first employs control point regression to establish sparse correspondences between distorted documents and their rectified counterparts, followed by interpolation-based dense mapping for geometric correction. Subsequently, a global attention mechanism predicts illumination parameters to normalize lighting variations in the rectified images. The key contributions include the following. First, we integrate a parameter-free joint attention mechanism for document rectification tasks, which significantly improves the distribution accuracy of control points along document edges through spatial–channel dual-dimensional energy optimization. Second, our architecture introduces a hybrid attention cascade after deep feature extraction, combining channel recalibration with spatial self-attention to amplify deformation-critical patterns; channel attention dynamically scales features through inter-channel dependencies, while spatial attention prioritizes regions exhibiting high surface curvature or folding artifacts. Finally, we introduce a novel two-stage framework combining improved geometric correction via control point regression with subsequent illumination normalization through attention-based illumination correction. Our method is validated using the DRIC [11] and Fiducial [5] datasets, which provide paired real-world distorted images and synthetic warping ground truths. Comprehensive evaluations on the standardized benchmark proposed in [4] demonstrate state-of-the-art performance across multiple metrics.

The remainder of this paper is organized as follows: Section 2 reviews related work on geometric unwarping and illumination correction for document images; Section 3 describes the datasets used in our study; Section 4 details our proposed DocCPLNet framework, including the overall architecture, geometric unwarping network, and illumination correction network; Section 5 presents experimental results validating the effectiveness of our approach, encompassing quantitative and qualitative comparisons with state-of-the-art methods along with ablation studies; finally, Section 6 concludes the paper and discusses potential future research directions.

## 2. Related Work

### 2.1. Geometric Unwarping

Recent years have witnessed significant advancements in geometric rectification techniques for distorted document images. Prior to the deep learning era, conventional methods primarily relied on extracting low-level features to construct 2D/3D rectification models. These approaches typically utilized visual cues such as document boundaries [14], text lines [15], and texture flows [16] to reconstruct document surfaces. The advent of deep learning has revolutionized this field by enabling data-driven approaches that learn distortion patterns directly from document images. Ma et al. [4] proposed a stacked U-Net architecture to predict forward mapping fields for document unwarping, establishing an end-to-end learning paradigm. Das et al. [7] proposed a two-stage deep learning framework for document image rectification. Their method first employs a shape network to predict a 3D coordinate map of the warped document, followed by a texture mapping network that generates a backward mapping field. The final rectified image is obtained through bilinear sampling based on the predicted mapping field. Feng et al. [17] proposed an approach that detects text lines in document images to encode rectification cues for improved geometric correction. Their method models detected text lines as cubic B-splines to guide the document unwarping process. Xue et al. [8] developed a Fourier-space transformation method that separates high-frequency text content from low-frequency background noise, enabling robust rectification without dense 3D annotations. Xie et al. [5] presented a computationally efficient solution using estimated control points and reference point interpolation, offering flexibility in handling various distortion types through adjustable point density. Feng et al. [12] introduced a transformer-based architecture for document image rectification, adapting the transformer model from natural language processing tasks to enhance representation learning for document images.

### 2.2. Document Illumination Correction

Document illumination correction is a critical sub-task of document rectification that aims to address photometric distortions caused by uneven lighting conditions. Early research primarily focused on leveraging handcrafted geometric and photometric features to model illumination effects. For instance, Ref. [10] utilized shading information derived from Shape-from-Shading to guide geometric unwarping while partially mitigating illumination inconsistencies. Ref. [9] proposed an illumination separation framework to decompose shading and reflectance components through optimization. Subsequent studies revealed that removing exposure variations and shadows significantly enhances Optical Character Recognition (OCR) accuracy, prompting dedicated efforts in this direction. Recent advances have shifted towards deep learning due to its capacity to learn complex illumination patterns. Ref. [11] introduced residual blocks with skip connections to balance local and global illumination variations, effectively normalizing document surfaces. Ref. [7] developed a dual U-Net architecture in which one network predicts surface normals and the other estimates shadow maps, which jointly addresses shading artifacts. Ref. [11] proposed a light-guided framework that predicts background illumination characteristics and enforces cycle consistency to correct spatially varying shadows. More recently, transformer-based methods have emerged; Ref. [12] designed a Vision Transformer to model long-range dependencies in illumination distortion, achieving state-of-the-art performance in both geometric and photometric restoration. Additionally, Ref. [13] explored unpaired learning with UDoc-GAN, leveraging background light priors and adversarial training to handle scenarios lacking paired data. These studies collectively demonstrate that deep learning provides a robust paradigm for illumination prediction, enabling end-to-end solutions that surpass traditional feature engineering approaches.

## 3. Dataset

To evaluate the proposed method, we employ three distinct datasets corresponding to different components of our framework. Specifically, the geometric correction network is trained on the synthetic Fiducial dataset proposed by Xie et al. [5], while the illumination prediction network utilizes the DRIC dataset introduced by Li et al. [11]. Final performance evaluation is conducted on the DocUNet benchmark [4], a widely recognized standard for document unwarping tasks. Below, we detail the characteristics and roles of these datasets.

The Fiducial dataset [5] comprises synthetically generated document images created through 2D mesh perturbations, simulating diverse geometric distortions such as folds and curves. This dataset provides precise ground-truth deformation fields, enabling supervised training of geometric correction models. For illumination modeling, the DRIC dataset [11] offers 3D-rendered document images with realistic lighting variations, including directional shadows and HDR environment maps, allowing networks to be trained in handling photometric artifacts. Finally, the DocUNet benchmark [4] contains 130 real-world document images captured under uncontrolled conditions, covering complex distortions such as crumpling, multi-view perspectives, and mixed text–graphics layouts. This benchmark serves as the evaluation standard, with scanned ground-truth images facilitating quantitative assessment via metrics such as MS-SSIM and Local Distortion (LD). Together, these datasets ensure comprehensive validation of our method’s geometric rectification and illumination normalization capabilities. Figure 1 illustrates representative samples from three benchmark datasets, covering real-world deformations, synthetic geometric perturbations, and illumination variations.

## 4. Methodology

### 4.1. Architecture Overview

While the control point-based DDCP method [5] achieves remarkable geometric rectification performance, it exhibits limitations when handling documents with complex backgrounds or non-uniform illumination. These challenges often lead to text distortion, residual background artifacts, and shadow retention, ultimately degrading downstream applications such as OCR. To address these issues, we propose DocCPLNet, a novel dual-branch architecture illustrated in Figure 2. Inspired by the hierarchical feature learning paradigm in [12], DocCPLNet integrates an Attention-Enhanced ATCP Network for robust geometric correction and a Global-Attention Transformer for illumination normalization. The ATCP network employs a spatial–channel hybrid attention mechanism to prioritize document edges and local deformation patterns, enabling precise control point prediction even under cluttered backgrounds. Subsequently, the illumination correction branch leverages transformer-based global attention to model long-range dependencies across the rectified document, effectively eliminating uneven lighting and residual shadows while preserving textural details. Compared to DDCP [5], our method achieves higher accuracy in control point regression by mitigating interference from background noise. Furthermore, the introduced illumination correction module addresses the critical limitation of DDCP in handling photometric distortions by utilizing adaptive shadow suppression and brightness homogenization. This dual optimization ensures enhanced readability and robustness for real-world document digitization scenarios.

The ATCP network utilizes a convolutional backbone with filters [32,32,64,128,256,256], while the GIT employs a standard transformer architecture with a model dimension $d_{\mathrm{model}}=512$, 6 encoder/decoder layers, and 8 attention heads, making the entire framework computationally efficient.

### 4.2. Geometric Unwarping Network

As illustrated in Figure 2 and Figure 3, our geometric rectification framework employs an ATCP Network to address document distortions. The ATCP network processes a distorted input image $I\in R^{992\times 992\times 3}$ through four cascaded submodules to predict control points $P\in R^{2\times 31\times 31}$, which correspond to reference points on a regular grid. The first submodule utilizes two convolutional layers with 3 × 3 kernels and a stride of 2 for initial feature extraction, followed by a joint attention mechanism inspired by [18]. This parameter-free attention module dynamically amplifies geometrically salient features along document edges by optimizing an energy function across spatial and channel dimensions. The joint attention mechanism enhances feature discrimination without introducing additional parameters, enabling precise localization of control points even under cluttered backgrounds.The second submodule performs deep feature extraction using four 3 × 3 convolutional layers, augmented by a hybrid attention mechanism from [19]. This module combines channel-wise recalibration and spatial self-attention to prioritize local deformation patterns. The channel attention adaptively scales feature responses based on inter-channel dependencies, while the spatial attention highlights regions with high curvature or folding artifacts. This dual attention strategy improves the network’s capacity to capture nonlinear distortions while suppressing noise. The third submodule uses a six-level dilated spatial pyramid. By hierarchically aggregating multi-scale contextual features, this pyramidal structure resolves ambiguities in large-scale deformations caused by perspective distortion or page curvature. The final submodule employs a two-layer convolutional network to regress control point coordinates. Each point $p_{i}=\left(x,y\right)$ on the 31 × 31 grid corresponds to a deformation-sensitive region, ensuring dense coverage of document boundaries and text lines.

Loss Function Design. Following Xie et al. [5], we adopt a multi-task loss combining several objectives. We define the necessary variables first, as follows: let the predicted and ground-truth control points be represented as sets of 2D coordinates $P=\{p_{1},p_{2},\ldots ,p_{N_{c}}\}$ and $R=\{r_{1},r_{2},\ldots ,r_{N_{c}}\}$, respectively, where $N_{c}$ is the total number of control points (31×31=961 in our setting). The reference points are fixed and denoted as $Q=\{q_{1},q_{2},\ldots ,q_{N_{c}}\}$, which form a uniform grid. The i-th control point $p_{i}$ corresponds to the i-th reference point $q_{i}$.

Smooth L1 Loss. For control point regression, this measures the distance between each predicted control point and its ground-truth counterpart, as shown below.(1)$$L_{\mathrm{smoothL}1}=\frac{1}{N_{c}}\sum_{i=1}^{N_{c}}\left\{\begin{array}{cc} 0.5{\left(p_{i}-{\hat{p}}_{i}\right)}^{2}, & \mathrm{if}\;|p_{i}-{\hat{p}}_{i}|\;<1\\ |p_{i}-{\hat{p}}_{i}|-0.5, & \mathrm{otherwise} \end{array}\right.$$

Differential Coordinate Constraints. These enforce the local smoothness of the predicted control point mesh. For each point $p_{i}$, we compute its differential coordinate $\delta_{i}$ as the sum of vectors from $p_{i}$ to its k nearest neighbors $p_{j}$, where j belongs to the neighborhood N(i) of $p_{i}$. The loss term $L_{c}$ is then defined as the mean squared error between the predicted ($\delta_{i}$) and ground truth (${\hat{\delta }}_{i}$) differential coordinates, as follows.(2)$$\begin{array}{c} L_{c}=\frac{1}{N_{c}}\sum_{i=1}^{N_{c}}{∥\delta_{i}-{\hat{\delta }}_{i}∥}_{2}^{2} \end{array}$$(3)$$\begin{array}{c} \delta_{i}=\sum_{j\in N(i)}\left(p_{j}-p_{i}\right) \end{array}$$

L1 Loss. Used for regulating the intervals between adjacent reference points, which helps to maintain a regular grid structure. We calculate the Euclidean distance $d_{i}$ between adjacent reference points in the ground-truth mesh. The loss $L_{r}$ is computed as the mean absolute error between these distances and the expected unit distances $\hat{d_{i}}$:(4)$$L_{r}=\frac{1}{N_{r}}\sum_{i=1}^{N_{r}}\left|d_{i}-\hat{d_{i}}\right|\;$$
where $N_{r}$ denotes the total number of edges between adjacent reference points considered in this calculation. The total loss is $L=L_{\mathrm{smoothL}1\;}+\alpha L_{c}+\beta L_{r}$, where α=0.1,β=0.01 balance the respective contributions. The ATCP network was optimized using the Adam optimizer with an initial learning rate of $2\times 10^{-4}$ and a batch size of 16. The learning rate was halved every 40 epochs throughout the 300-epoch training schedule on the Fiducial dataset. The weight decay was set to $1\times 10^{-4}$ for regularization.

### 4.3. Illumination Correction Network

Following geometric rectification, residual illumination artifacts are addressed using a Global Illumination Transformer (GIT), adapted from the architecture in Feng et al. [12]. As shown in Figure 4, the GIT employs a transformer encoder–decoder structure to model long-range dependencies across the document surface, disentangling illumination variations from content features. The geometrically unwarped image $I_{R}\in R^{H\times W\times 3}$ is partitioned into overlapping patches of size 128×128; these are flattened into sequences of $N_{p}=\frac{H\times W}{P^{2}}$ tokens, where P=16. Each token is linearly projected into a d-dimensional embedding d=512 and augmented with learnable positional encodings. This captures global illumination patterns by computing the attention scores between all token pairs:(5)$$\mathrm{Attention}\left(Q,K,V\right)=\mathrm{Softmax}\left(\frac{{\mathrm{QK}}^{T}}{\sqrt{d_{k}}}\right)V$$
where Q,K,V are the query, key, and value matrices derived from the input embeddings. Learnable illumination queries $Q\in R^{N_{q}\times d}\left(N_{q}=256\right)$ are utilized to aggregate the global context from the encoder outputs via cross-attention. Local brightness adjustments are iteratively refined while preserving textural sharpness through residual connections and layer normalization. To mitigate blurring artifacts from conventional upsampling, GIT incorporates a lightweight convolutional tail network. The decoded features are reshaped into $\frac{H}{8}\times \frac{W}{8}\times d$ and upsampled via transposed convolutions with learned adaptive filters, followed by a final convolution to predict the illumination-corrected output.

The network is optimized using a composite loss:(6)$$L_{\mathrm{illumination}}=\lambda_{1}{∥I_{\mathrm{GT}}-I_{C}∥}_{1}+\lambda_{2}L_{\mathrm{VGG}}\left(I_{\mathrm{GT}},I_{C}\right)$$
where $L_{\mathrm{VGG}}$ computes the $L_{1}$ distance between VGG-19 features extracted from multiple layers to enforce perceptual similarity. The GIT was trained using the AdamW optimizer with a stronger weight decay of 0.01 and an initial learning rate of $1\times 10^{-4}$. The learning rate was reduced by a factor of 0.25 when the validation loss plateaued after a patience period of 20 epochs. Training proceeded for 50 epochs on the DRIC dataset with a batch size of 16. The loss weights were set to $\lambda_{1}=1.0$ and $\lambda_{2}=10^{-5}$.

## 5. Experiments

### 5.1. Evaluation Metrics

The goal of document rectification is to enhance character recognition accuracy and document readability. A distorted document exhibits improved readability when the rectified output closely resembles its scanned counterpart. To quantitatively evaluate the proximity between processed and ground-truth scanned images, we employ MS-SSIM [20] and LD [21], which measure perceptual and pixel-level alignment while tolerating minor spatial discrepancies. For assessing OCR performance, ED and CER [22] are adopted as key metrics to quantify text recognition accuracy through character-level discrepancies.

MS-SSIM [20]: The Multiscale Structural Similarity evaluates perceptual quality by comparing luminance, contrast, and structural features across multiple resolution scales. Unlike single-scale SSIM, it accounts for viewing condition variations, making it ideal for assessing multi-resolution geometric preservation in document images. We compute MS-SSIM between rectified and ground-truth scanned images to quantify structural restoration, with higher values indicating superior alignment. Given two images x and y, MS-SSIM is defined as follows:(7)$$\mathrm{MS-}\mathrm{SSIM}\left(x,y\right)={\left[l_{M}\left(x,y\right)\right]}^{\alpha_{M}}\prod_{j=1}^{M}{\left[c_{j}\left(x,y\right)\right]}^{\beta_{j}}{\left[s_{j}\left(x,y\right)\right]}^{\gamma_{j}}$$
where $l_{M}$, $c_{j}$, and $s_{j}$ represent the luminance, contrast, and structure comparisons at scale j, respectively, and $\alpha_{M},\beta_{j},\gamma_{j}$ are parameters to adjust the relative importance of each component. Higher values indicate superior structural alignment.

LD [21]: The LD is a shift-invariant loss that measures pixel-level discrepancies while tolerating minor spatial misalignments between images. By incorporating a distance transform, the LD mitigates sensitivity to registration errors caused by residual local distortions. This metric ensures robust evaluation of global shape recovery, particularly for documents with complex curvature or folds. For images $I_{\mathrm{rec}}$ (rectified) and $I_{\mathrm{GT}}$ (ground truth), the LD is computed as follows:(8)$$\mathrm{LD}\left(I_{\mathrm{rec}},I_{\mathrm{GT}}\right)=\frac{1}{N}\sum_{p}\min_{q\in N(p)}{∥I_{\mathrm{rec}}\left(p\right)-I_{\mathrm{GT}}\left(q\right)∥}_{2}$$
where N(p) denotes a local neighborhood around pixel p and N is the total number of pixels. Lower LD values indicate better geometric restoration.

ED: The Edit Distance quantifies the minimum insertions, deletions, or substitutions required to transform OCR-recognized text into the ground truth. As a Levenshtein distance-based metric, the ED captures cumulative character-level discrepancies, directly reflecting the impact of geometric distortions on OCR reliability. Lower ED values indicate improved legibility and reduced segmentation errors in rectified documents. The ED is defined recursively as follows:(9)$$\mathrm{ED}\left(i,j\right)=\left\{\begin{array}{cc} \max(i,j) & \mathrm{if}\;\min(i,j)=0\\ \min\left\{\begin{array}{c} \mathrm{ED}(i-1,j)+1\\ \mathrm{ED}(i,j-1)+1\\ \mathrm{ED}\left(i-1,j-1\right)+1_{S_{\mathrm{rec}}\left[i\right]\neq S_{\mathrm{GT}}\left[j\right]} \end{array}\right. & \mathrm{otherwise} \end{array}\right.$$
where 1 is the indicator function. Lower ED values indicate improved legibility.

CER [22]: The Character Error Rate normalizes the ED by the total number of characters in the reference text, providing an interpretable measure of OCR accuracy. The CER is critical for evaluating text recognition performance on warped documents, where local distortions degrade stroke continuity. Our method minimizes the CER by prioritizing spatial and channel features that enhance character-level structural integrity. The CER normalizes the ED by the total number of characters $N_{\mathrm{GT}}$ in the ground truth text:(10)$$\mathrm{CER}=\frac{\mathrm{ED}(S_{\mathrm{rec}},S_{\mathrm{GT}})}{N_{\mathrm{GT}}}.$$

The CER provides an interpretable measure of OCR accuracy, with lower values indicating better performance.

To address potential variability in test results across different images and models, we standardized our evaluation using the methodology proposed in [4,12] when conducting experiments on the publicly available DocUNet benchmark [4]. This protocol ensures unbiased comparisons by aggregating results across diverse distortion types and imaging conditions, validating the robustness and generalizability of our approach. Figure 5 illustrates the different methods used to evaluate the rectified images.

### 5.2. Implementation Details

We implemented our DocCPLNet using PyTorch 1.5.1 [23]. The geometric correction network (ATCP) and global illumination prediction network (GIT) were trained on the Fiducial dataset [5] and DRIC dataset [11], respectively.

Geometric Correction Network (ATCP) This network was trained on synthetic data generated by warping scanned documents using 2D mesh transformations. The documents were augmented with random shadows, affine transformations, Gaussian blur, and background textures. We adopted a Smooth L1 loss combined with differential coordinate constraints weighted by α=0.1 to enforce local positional correlations between control points. The Adam optimizer was utilized with an initial learning rate of $2\times 10^{-4}$, which was reduced by half every 40 epochs. Training spanned 300 epochs with a batch size of 16.

Global Illumination Prediction Network (GIT): Inspired by [12], the GIT leverages a transformer-based encoder–decoder architecture to address illumination artifacts. The network processes 128×128 image patches cropped from geometrically rectified outputs, with a 12.5% overlap to ensure seamless stitching. Training on the DRIC dataset [11] employed a hybrid loss combining L1 reconstruction loss and VGG perceptual loss weighted by $\alpha =10^{-5}$ to preserve high-frequency details. We used the AdamW optimizer with an initial learning rate of $1\times 10^{-4}$, which was decayed by a factor of 0.25 after 20 epochs. The model was trained for 50 epochs with a batch size of 16, incorporating random HSV jitter and Gaussian noise augmentation to improve robustness.

### 5.3. Experimental Results

To validate the effectiveness of DocCPLNet, we conducted comprehensive experiments on the DocUNet benchmark [4] comparing it against seven state-of-the-art learning-based methods. All tests were performed on an RTX 4060 GPU (Windows 11) with Tesseract 5.0.1 [24] for OCR accuracy evaluation. Quantitative results are summarized in Table 1, while Figure 6 illustrates qualitative improvements in geometric rectification and illumination harmonization.

Our method achieves a CER of 0.180, surpassing all non-transformer baselines (e.g., DewarpNet [7], 0.230; DDCP [5], 0.340) and closely approaching the transformer-based GeoTr [17] (0.183). While certain methods achieve excellent performance on structural similarity, notably the transformer-based DocTr [12] and GeoTr [17], a key advantage of our DocCPLNet lies in its favorable balance between accuracy and computational efficiency. The architectural design of our Attention-Enhanced ATCP Network, which is based on a convolutional backbone with efficient hybrid attention modules, is inherently more lightweight than the full self-attention mechanisms used in transformer models, which have quadratic complexity with respect to input size. This design choice results in a significantly lower computational footprint and faster inference times, making DocCPLNet a more practical and deployable solution for real-world applications on resource-constrained devices such as mobile phones without sacrificing competitive OCR performance. This highlights the synergy between our ATCP and GIT networks.

To further demonstrate the generalization capabilities of DocCPLNet, we evaluated its performance on both the DIR300 [17] test set and out-of-distribution samples, including historical manuscripts and print forms. As illustrated in Figure 7, our method effectively rectifies challenging document distortions such as incomplete pages and locally warped text regions while maintaining superior geometric fidelity compared to existing learning-based approaches. For instance, in cases where documents exhibit non-uniform folding or partial occlusion, DocCPLNet minimizes residual distortions while preserving text continuity, as evidenced by the reduced CER (0.180).

Remarkably, even without explicit training on specialized datasets, DocCPLNet achieves a CER of 0.212 on out-of-distribution samples such as historical manuscripts. This robustness stems from two key innovations:

ATCP’s Differential Coordinate Constraints: By enforcing geometric consistency through sparse control point regularization, the network adapts to arbitrary document layouts and content types without overfitting to specific distortion patterns.

GIT’s Dynamic Illumination Augmentation: Simulating diverse lighting conditions during training enables robust handling of shadowed or unevenly lit text regions in real-world scenarios. The qualitative results in Figure 7 further reveal that DocCPLNet avoids common artifacts observed in prior methods, such as blurred text boundaries [11] or over-stretched paragraphs [7]. Quantitative metrics and visual comparisons collectively confirm that our framework achieves strong generative capability in restoring diverse real-world document images while maintaining practical deployability on mid-tier hardware.

The performance of our proposed DocCPLNet remains lower than that of state-of-the-art transformer-based methods such as DocTr [12], particularly on distortion metrics such as MS-SSIM. We attribute this performance gap primarily to the fundamental architectural differences between the two models. The self-attention mechanism of transformers excels at modeling long-range dependencies across the entire image, which is highly beneficial for capturing global document structure and illumination fields. In contrast, the convolutional backbone of our geometric correction branch has a more limited receptive field and global modeling capacity. This represents a inherent tradeoff between the high performance of large-capacity models and the practical efficiency of our more compact design. Our objective was to develop a robust and efficient framework that provides competitive performance with significantly lower computational overhead.

### 5.4. Experimental Ablation

To validate the effectiveness of key components in our framework, we conducted ablation studies on the DocUNet benchmark [4], comparing three variants: (a) DDCP [5] with joint attention mechanisms, (b) our proposed Attention-Driven Thin-Plate Control Points Network, and (c) the full DocCPLNet integrating ATCP and the Global Illumination Transformer. Quantitative results are summarized in Table 2, with qualitative comparisons shown in Figure 8. As shown in Table 2, variant (a) achieves an MS-SSIM of 0.476 and CER of 0.25, already surpassing the original DDCP [5] by leveraging joint attention for feature correlation. However, its limited geometric modeling capability results in high ED and LD, indicating residual distortions in text line topology. Replacing the baseline control point network with our ATCP reduces the ED by 21.3% and CER by 16%, demonstrating the effectiveness of the hybrid attention mechanism in balancing local deformation modeling and global structural constraints. The slight drop in MS-SSIM reflects ATCP’s prioritization of OCR-compatible rectification over absolute pixel-level similarity.

Role of GIT Module: The full DocCPLNet achieves optimal performance across all metrics, proving that GIT’s illumination harmonization complements ATCP’s geometric correction. Specifically, GIT reduces illumination-induced OCR errors by 14.3% while maintaining geometric fidelity. GIT effectively suppresses shadow artifacts and enhances text contrast in low-light regions, which directly contributes to the robustness of downstream OCR systems.

Tradeoff Analysis: Notably, the LD metric increases slightly from 8.541 to 8.687 in the full model, suggesting that global illumination correction may introduce minor local deformations. However, this tradeoff is justified by the significant CER reduction and ED improvement, aligning with our design goal of optimizing for practical document digitization workflows.

The ablation results validate our architectural design choices and highlight the necessity of co-optimizing geometric and photometric corrections for real-world document digitization systems.

## 6. Conclusions

This paper proposes DocCPLNet, a unified framework for document image restoration that jointly addresses geometric deformation and illumination distortion. Our approach synergistically integrates an attention-driven control point prediction network for geometric rectification with a global illumination transformer for photometric correction. The key benefits of our method are threefold. (1) Enhanced Geometric Accuracy: The ATCP module equipped with hybrid spatial–channel attention mechanisms achieves precise localization of control points, leading to superior rectification of complex distortions such as folds and curls while preserving textual integrity. (2) Robust Illumination Normalization: The subsequent GIT module effectively models long-range dependencies to eliminate shadows and uneven lighting, significantly improving contrast and readability. (3) Superior Downstream Performance: This dual-branch design works cohesively to produce rectified images that are not only visually pleasing but also optimized for practical applications, as evidenced by our state-of-the-art OCR accuracy on the DocUNet benchmark.

Despite its strong performance, the proposed method has certain limitations that warrant discussion. While the transformer-based GIT module is effective, its computational complexity is higher than simpler convolutional alternatives, which could be a constraint for real-time applications on resource-constrained devices. Our framework’s performance is inherently tied to the quality and diversity of the training data. While we used established synthetic datasets, namely, Fiducial and DRIC, the model may still struggle with extreme real-world distortions or lighting conditions that are not well-represented during training, such as heavy occlusion or highly specular reflection. The current pipeline is a sequential two-stage process involving geometry followed by illumination. As such, an error in the first geometric stage could propagate and be amplified in the second illumination stage, and the framework does not explicitly model potential interactions between geometric warp and illumination artifacts.

Future work will focus on addressing these drawbacks and exploring several promising improvements. To mitigate computational costs, we will investigate knowledge distillation or network pruning techniques to create a more lightweight version of the GIT module without significant performance loss. To enhance generalization, we plan to incorporate semi-supervised or self-supervised learning strategies utilizing unlabeled real-world document image in order to reduce the dependency on perfectly paired synthetic data. Furthermore, we will explore an end-to-end jointly trained architecture that allows for iterative refinement between geometric and photometric corrections, potentially leading to more robust and cohesive restoration. Finally, we will focus on refining text-line and character-level feature representation within the ATCP network to better handle irregular layouts and degraded historical documents. This direction aligns with emerging VR/AR digitization demands, where hybrid sensing technologies could enable real-time 3D document reconstruction and interactive restoration.

## Figures and Tables

**Figure 1 sensors-25-06304-f001:**
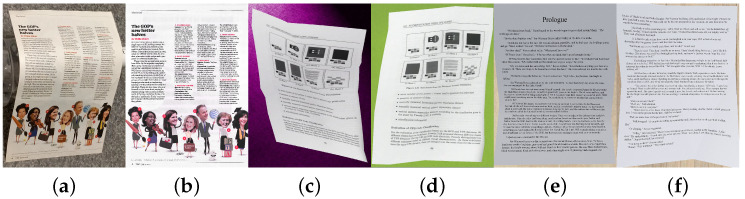
(**a**,**b**) DocUNet [4]; (**c**,**d**) Fiducial [5]; (**e**,**f**) DRIC [11].

**Figure 2 sensors-25-06304-f002:**
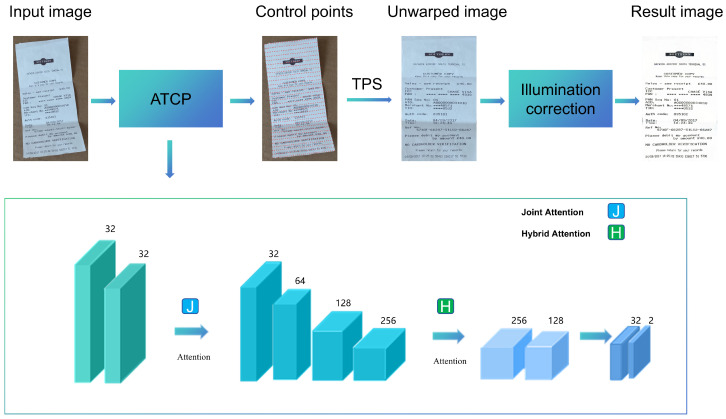
Complete architecture.

**Figure 3 sensors-25-06304-f003:**
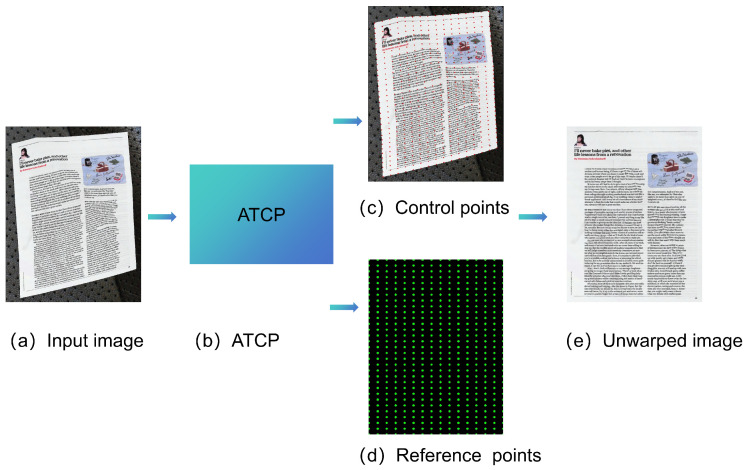
Geometric unwarping network.

**Figure 4 sensors-25-06304-f004:**
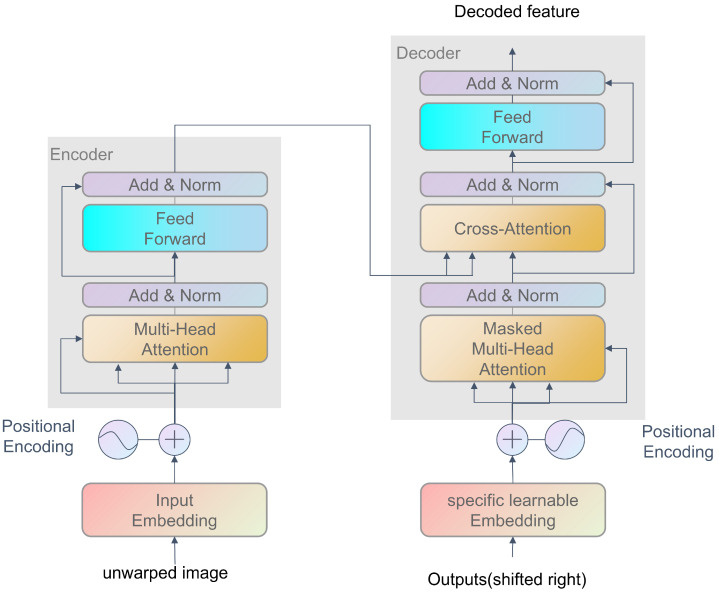
Architecture of the illumination correction transformer.

**Figure 5 sensors-25-06304-f005:**
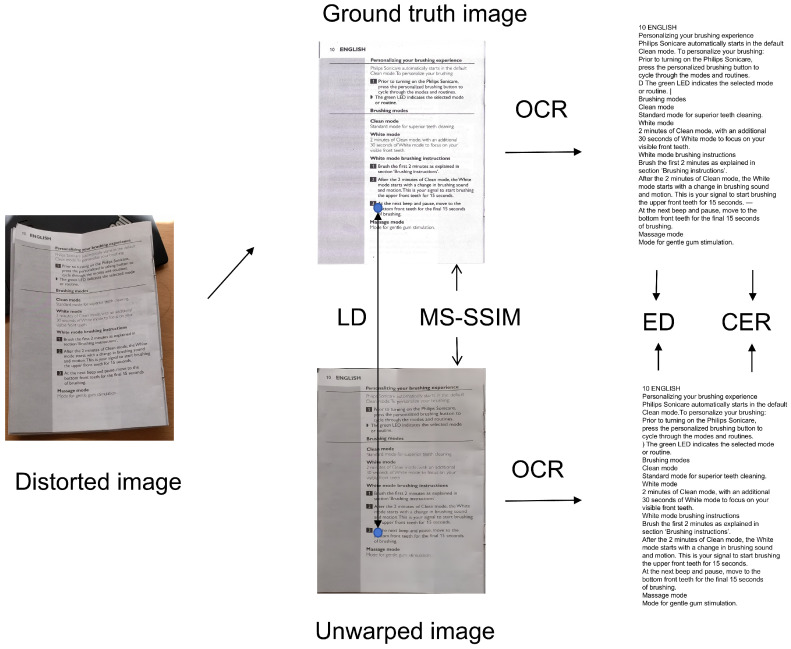
Quantitative and qualitative comparisons of rectification methods.

**Figure 6 sensors-25-06304-f006:**
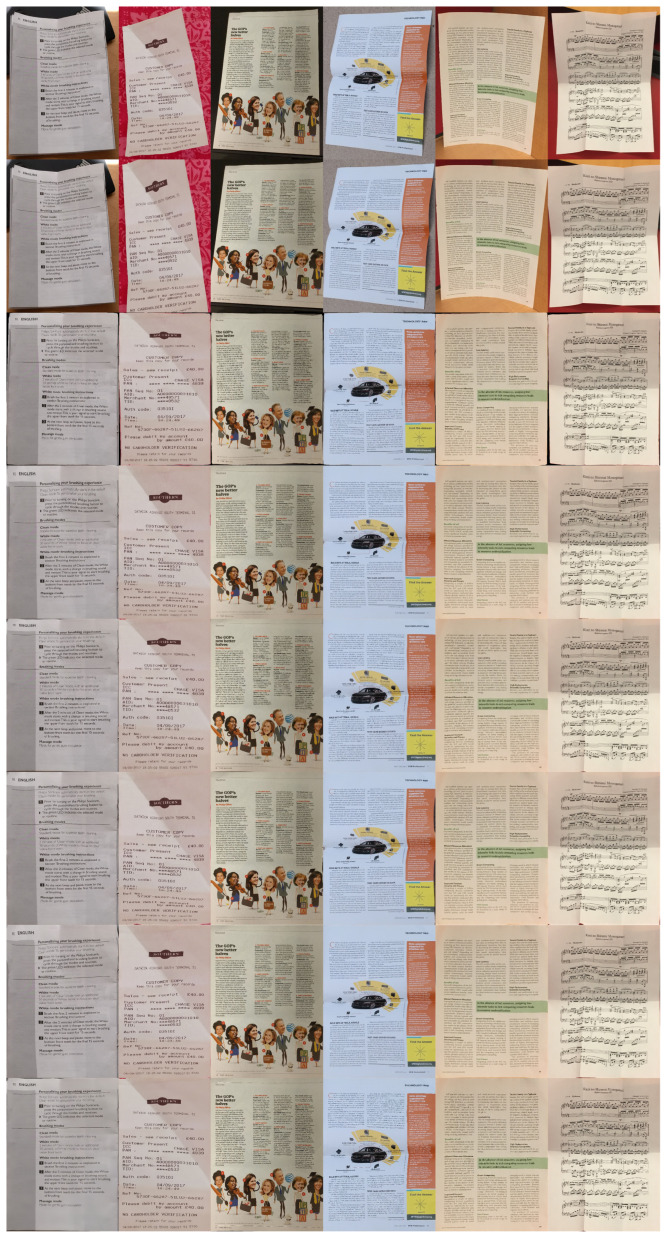
Quantitative comparisons on the DocUNet [4] benchmark with existing learning-based methods, including DocProj [11], DewarpFlow [25], Doctr [12], DDCP [5], DewarpNet [7], Geotr [17], and our Geometric Unwarping Network (ATCP).

**Figure 7 sensors-25-06304-f007:**
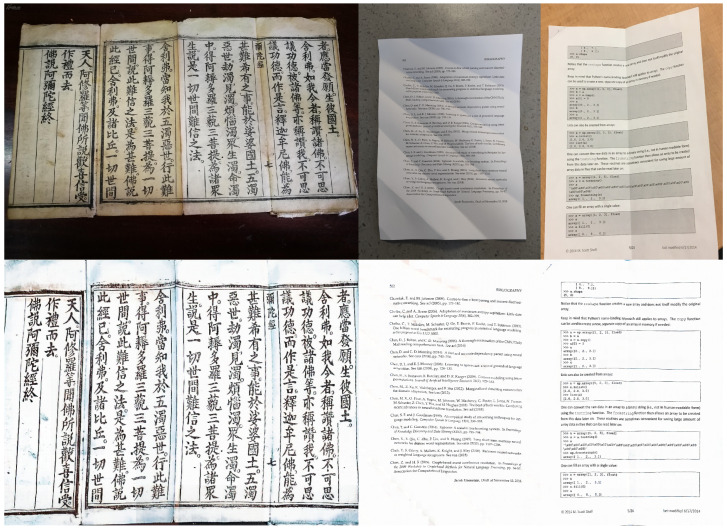
Correction results on the DIR300 [16] test set and handwritten manuscripts.

**Figure 8 sensors-25-06304-f008:**
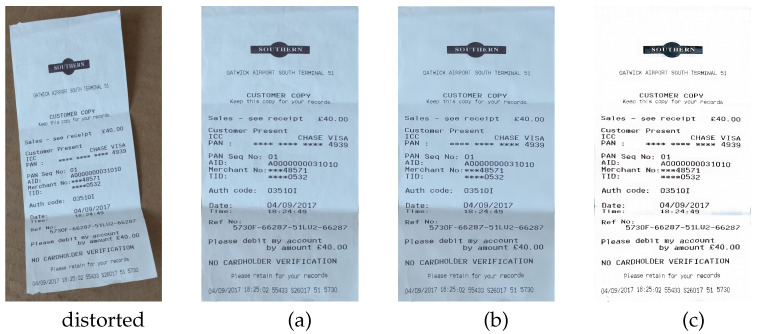
Qualitative comparison with ablation methods (**a**), (**b**), and (**c**).

**Table 1 sensors-25-06304-t001:** Quantitative comparisons with existing methods on the DocUNet benchmark dataset [3]. “↑” indicates that higher is better, while “↓” means the opposite.

Methods	MS-SSIM ↑	LD ↓	ED ↓	CER ↓
Distored	0.246	20.507	2789.1	0.613
DocUNet [4]	0.413	14.193	1239.12	0.384
DocProj [11]	0.272	19.51	1165.93	0.3818
Doctr [12]	0.496	8.014	339.59	0.1164
DDCP [5]	0.475	9.106	1248.06	0.34
Geotr [17]	0.502	8.287	592.29	0.183
DewarpFlow [25]	0.428	7.771	1260.83	0.35
DewarpNet [7]	0.4693	8.987	744.77	0.23
DocCPLNet	0.477	8.687	594.89	0.180

**Table 2 sensors-25-06304-t002:** Ablation experiments on DocCPLNet. The setting used in our final model is underlined. “↑” indicates that higher is better, while “↓” means the opposite.

	DocUNet		
Methods	(a)	(b)	(c)
+Joint Attention	✓	✓	✓
+Hybrid Attention		✓	✓
+GIT			✓
MS-SSIM ↑	0.476	0.472	0.477
LD ↓	9.043	8.541	8.687
ED ↓	958.05	754.16	594.89
CER ↓	0.25	0.21	0.18

## Data Availability

The fiducial1024 dataset is made publicly available for research purposes. For more information, please refer to the website: https://github.com/gwxie/Document-Dewarping-with-Control-Points (accessed on 25 September 2025). The DRIC dataset is made publicly available for research purposes. For more information, please refer to the website: https://github.com/xiaoyu258/DocProj (accessed on 25 September 2025). The DocUNet dataset is made publicly available for research purposes. For more information, please refer to the website: https://www3.cs.stonybrook.edu/~cvl/docunet.html (accessed on 25 September 2025). The DIR300 dataset is made publicly available for research purposes. For more information, please refer to the website: https://drive.google.com/drive/folders/1yySouQQ3BlH7OjnUhq4CLuvpX2KXtifX?usp=sharing (accessed on 25 September 2025).

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
