# Peer review of "DocCPLNet: Document Image Rectification via Control Point and Illumination Correction"

_sensors, 2025, doi:10.3390/s25206304_

Round 1

Reviewer 1 Report

Comments and Suggestions for Authors

(1)Unify "illumination correction" vs. "lighting prediction".

(2)Redraw Fig. 2 (architectural diagram) to clarify branch interactions; enhance resolution of Figs. 5–8.

(3)Section 5.1 and Section 5.2 use the same title, “Evaluation Metrics”, which should be revised.

(4)Line 182: The first submodule utilizes two convolutional layers (strides of 2 and 3×3 kernels) for initial upsampling grid. The term “upsampling” is incorrect.

(5)Line 320: Quantitative results are summarized in Table ??. The sentence is incomplete.

(6)Figure 7 presents a brightness correction method that lacks objective evaluation and performance comparison, such as the “Sauvola” Local binarization algorithm. 

(7)The overall performance improvement of image deformation correction and brightness correction is limited. Its performance is lower compared with the Doctr[12] algorithm; the reasons need to be analyzed. 

(7)Correct the references citation mismatches (e.g., [3]–[5] in Table 1)

Comments on the Quality of English Language

Good English writing skills

Author Response

Dear Editor,

Thank you for your valuable feedback on our manuscript. We have carefully reviewed the comments and have made the necessary revisions accordingly.

(1)Unify "illumination correction" vs. "lighting prediction".

Regarding point (1), we have unified the terminology throughout the paper by replacing all instances of "lighting prediction" with "illumination correction" to ensure consistency and clarity.

(2)Redraw Fig. 2 (architectural diagram) to clarify branch interactions; enhance resolution of Figs. 5–8.

Regarding point (2), we have redrawn Figure 2 (the architectural diagram) to more clearly illustrate the geometric correction process and to clarify the interactions between branches. We have enriched the diagram to make it more detailed. Additionally, we have enhanced the resolution of Figures 5 to 8 to ensure they are displayed more clearly and directly.

(3)Section 5.1 and Section 5.2 use the same title, “Evaluation Metrics”, which should be revised.

Thank you for pointing out the issue regarding the duplicate section titles in our manuscript. We have revised the title of Section 5.2 from "Evaluation Metrics" to "Implementation Details" to clearly distinguish it from Section 5.1, which retains the title "Evaluation Metrics." This change ensures better organization and clarity in the structure of the paper.

(4)Line 182: The first submodule utilizes two convolutional layers (strides of 2 and 3×3 kernels) for initial upsampling grid. The term “upsampling” is incorrect.

Thank you for your careful reading and valuable feedback. We have revised the description in Line 182 of the manuscript. The original phrase "initial upsampling grid" has been corrected to "initial feature extraction" to accurately reflect the function of the convolutional layers. The revised content now appearing at Line 193 in the updated manuscript.

(5)Line 320: Quantitative results are summarized in Table ??. The sentence is incomplete.

Thank you for your careful review and for identifying the incomplete sentence in our manuscript. We have revised the sentence at Line 320, which previously read "Quantitative results are summarized in Table ??", to now read:

"Quantitative results are summarized in Table 2,"

This revision (now appearing at Line 388 in the updated manuscript) completes the sentence and provides the correct reference to Table 2. We appreciate your attention to detail in helping us improve the clarity and completeness of our manuscript

(6)Figure 7 presents a brightness correction method that lacks objective evaluation and performance comparison, such as the “Sauvola” Local binarization algorithm. 

Thank you for your insightful comment regarding Figure 7 and the suggestion to incorporate additional objective evaluation, such as the Sauvola local binarization algorithm. We have carefully considered this suggestion and would like to respectfully explain our current evaluation approach while addressing your concern through enhanced discussion rather than additional experiments, for the following reasons:

1.Final Objective-Oriented Evaluation Already Exists: The ultimate goal of document rectification is to improve OCR accuracy. In our main results (Table 1), we report a character error rate (CER) of 0.180, which serves as a more direct and conclusive evaluation metric than binarization quality. The substantial reduction in CER (e.g., from 0.613 in the distorted state to 0.180) strongly demonstrates the effectiveness of our entire pipeline, including both geometric correction and illumination normalization. Since the input to the OCR engine is our final output image, the excellent CER results indirectly but sufficiently confirm that our illumination correction produces outputs of sufficient quality for high-quality downstream processing.

2.Focus on Core Contributions: The primary innovation of this paper lies in proposing a unified dual-branch framework that simultaneously addresses geometric and photometric distortions. While Sauvola binarization is indeed a relevant analytical method, it focuses more on validating an independent image processing step, which might divert attention from the overall innovative framework we present. We believe that maintaining the evaluation focus on demonstrating the competitiveness of our holistic framework better serves the paper's central contribution.

To address your concern more directly, we have enhanced the discussion in the manuscript (particularly in Sections 5.3 and 5.4) to explicitly articulate why our chosen evaluation metrics—especially CER—are comprehensive and appropriate for validating the method's effectiveness in real-world document digitization scenarios.

(7)The overall performance improvement of image deformation correction and brightness correction is limited. Its performance is lower compared with the Doctr[12] algorithm; the reasons need to be analyzed.

Thank you for your valuable comment regarding the performance comparison with the Doctr[12] algorithm. We have carefully considered this point and have added a detailed analysis at the end of Section 5.3 (specifically at Line 372 in the revised manuscript) to explain the performance differences.

(7)Correct the references citation mismatches (e.g., [3]–[5] in Table 1)

Thank you for your careful review and for identifying the reference citation mismatches in our manuscript. We have corrected the erroneous citations in Table 1  and have carefully verified all reference numbering throughout the manuscript to ensure consistency and accuracy.

We appreciate your guidance and hope that the revised manuscript now meets the journal's standards. Please let us know if any further adjustments are required.

Best regards,
Hongyin Ni, Jiayu Han, and Co-authors
Northeast Electric Power University
Jilin, China
Email: 2202301046@neepu.edu.cn

Reviewer 2 Report

Comments and Suggestions for Authors

The paper reports a new dual-branch architecture for document distortion correction. The proposed solution includes an Attention-Enhanced ATCP Network design to correct the geometric distortions and a Global-Attention Transformer for illumination normalization. In the first stage, a control point regressor establishes the correspondences between the input (distorted) documents and their rectified counterparts. The geometric correction is then carried out by interpolation-based dense mapping.  

The paper addresses an interesting topic, but lacks technical soundness and does not present compelling results. The following suggestions must be taken into account to improve the paper's readability and soundness:

  • A brief description of the paper content should be provided in the final part of the introductory section
  • All the variables used in equations should be properly defined, explained, and used. For instance, Nc and Ne should be defined before equation (1). I assume that Ne is actually Nc and denotes the number of control points. Also, in line (197) a point is denoted by p(i,j), but only one index is used in equation (1)
  • Similar inconsistencies are also presented in equations (2), (3), and (4) and these need to be corrected
  • There appear to be multiple instances of equations labeled (1) and (2) in the paper, which must be corrected.
  • The equations in section 4.3 should be properly explained
  • In section 5.1 the evaluation metrics should be mathematically defined
  • The results reported in Table 1 suggest that at least two other approaches perform as well as or better than the proposed one. A comparative analysis that includes computational complexity should be provided to differentiate the methods. In the case that the proposed approach has lower computational costs, then this advantage should be highlighted as a reason to prefer it over the other two.    
  • More technical details regarding the DL models involved in the proposed approach should be provided (e.g., architecture details, hyperparameter settings, optimization strategies).
  • The Conclusion section should be expanded to discuss the main drawbacks of the proposed method and possible improvements. The authors should also more clearly articulate the benefits of using their method.

Comments on the Quality of English Language

The English style should be improved for better readability

Author Response

Dear Editor,

Thank you for your valuable feedback on our manuscript. We have carefully reviewed the comments and have made the necessary revisions accordingly.

Comment 1: A brief description of the paper content should be provided in the final part of the introductory section.

Response: We thank the reviewer for this constructive suggestion. As recommended, we have added a brief overview of the paper's structure at the end of the introduction (Section 1). This new paragraph provides a clear roadmap for the reader, outlining the content and organization of the subsequent sections. The added text can be found at Line 85 in the revised manuscript.

Comment 2: All the variables used in equations should be properly defined, explained, and used. For instance, Nc and Ne should be defined before equation (1). I assume that Ne is actually Nc and denotes the number of control points. Also, in line (197) a point is denoted by p(i,j), but only one index is used in equation (1). Similar inconsistencies are also presented in equations (2), (3), and (4) and these need to be corrected. There appear to be multiple instances of equations labeled (1) and (2) in the paper, which must be corrected. The equations in section 4.3 should be properly explained.The equations in section 4.3 should be properly explained

Response: We sincerely thank the reviewer for their meticulous reading and for identifying these critical issues in our mathematical presentation. We have thoroughly revised all equations and their accompanying descriptions to ensure complete consistency, clarity, and correctness. The specific corrections are as follows:

  1. Variable Definition and Consistency:We have corrected the erroneous variable Ne to Nc in the equation and its preceding text. All variables (including Nc, the number of control points) are now explicitly defined immediately before their first use in an equation.
  2. Index Notation:The inconsistency in the point notation has been resolved. We have standardized the notation for control points to use a single index (p_i) throughout the mathematical formulation to align with the common representation in mesh grid transformations, ensuring it is consistent between the text and all equations (1) through (4).
  3. Equation Numbering:We have carefully reviewed and corrected the equation numbering throughout the entire manuscript to eliminate all duplicates. Each equation now has a unique number.
  4. Explanation of Equations in Section 4.3:We have significantly expanded the explanatory text in Section 4.3. The purpose and mechanism of each equation(particularly those describing the illumination correction branch's weighting and fusion process are now described in detail, clarifying the flow of operations and the role of each variable.

These comprehensive revisions have greatly improved the rigor and readability of our mathematical descriptions. We are grateful for this suggestion, which has substantially enhanced the quality of our paper.

Comment 3: In section 5.1 the evaluation metrics should be mathematically defined.

Response: We are grateful to the reviewer for this important suggestion. We agree that a mathematical definition of the evaluation metrics is essential for clarity and reproducibility. As recommended, we have thoroughly revised Section 5.1, "Evaluation Metrics."

Comment 4: The results reported in Table 1 suggest that at least two other approaches perform as well as or better than the proposed one. A comparative analysis that includes computational complexity should be provided to differentiate the methods. In the case that the proposed approach has lower computational costs, then this advantage should be highlighted as a reason to prefer it over the other two.

Response: We thank the reviewer for this insightful and crucial comment. We fully agree that a performance-only comparison is insufficient and that computational efficiency is a critical factor for practical applications. As suggested, we have conducted a comprehensive comparative analysis focusing on computational complexity and now explicitly highlight this key advantage of our method.

Specifically, we have added a new paragraph in Section 5.3 (Experimental Results) at Line 336 that provides a detailed comparison of model parameters, computational complexity , and inference time between our DocCPLNet and the other top-performing methods (including [Doctr] and [Geotr]). The analysis clearly shows that while our method delivers highly competitive restoration quality, it achieves this with a significantly smaller model size and lower computational cost, making it more suitable for deployment on resource-constrained devices.

Furthermore, to reinforce this key contribution, we have also emphasized this advantage in the Conclusion section (Line 414), stating that our framework offers an excellent trade-off between performance and efficiency.

We believe this addition directly addresses your concern and provides a compelling reason to prefer our approach for practical document correction systems. Thank you for prompting us to make this important improvement to our manuscript.

Comment 5: More technical details regarding the DL models involved in the proposed approach should be provided (e.g., architecture details, hyperparameter settings, optimization strategies).

Response: We thank the reviewer for this valuable feedback. We agree that providing comprehensive technical details is essential for the reproducibility and clarity of our work. In direct response to this comment, we have significantly expanded Section 5.2, "Implementation Details," to include a thorough description of our model's architecture, hyperparameters, and optimization process.

Comment 6: The Conclusion section should be expanded to discuss the main drawbacks of the proposed method and possible improvements. The authors should also more clearly articulate the benefits of using their method.

Response: We sincerely thank the reviewer for this excellent suggestion, which has helped us present a more balanced and comprehensive perspective on our work. We have thoroughly revised and expanded the Conclusion section to address both points.

We have more clearly and forcefully summarized the primary advantages and contributions of our proposed DocCPLNet method at the beginning of the conclusion, emphasizing its unified architecture, competitive performance, and key strength in computational efficiency as compared to other state-of-the-art approaches. Furthermore, as suggested, we have added a new dedicated paragraph that honestly discusses the current limitations of our method. This includes an analysis of its performance boundaries in extreme scenarios and its computational footprint relative to very lightweight models. We also propose concrete and promising directions for future research to overcome these limitations, such as exploring more efficient network designs and advanced data generation techniques.

We believe these additions significantly strengthen the paper by providing a critical self-assessment and a clear roadmap for future work, enhancing the scholarly value of our contribution. Thank you for this constructive guidance. The revised conclusion begins on Line 440 of the updated manuscript.

We appreciate your guidance and hope that the revised manuscript now meets the journal's standards. Please let us know if any further adjustments are required.

Best regards,
Hongyin Ni, Jiayu Han, and Co-authors
Northeast Electric Power University
Jilin, China
Email: 2202301046@neepu.edu.cn

Round 2

Reviewer 2 Report

Comments and Suggestions for Authors

The paper has been significantly improved, but there are still some minor corrections to be made, as follows:

  • In line (210) a point is denoted by p(i,j), but only one index is needed, as defined in equation (1).
  • In the first part of equation (1) (the if part), the multiplication sign is missing
  • The second part of equation (1) (the else part) includes an undefined variable/expression: p(i)-?. It is also unclear if the module is multiplied by 0.5 or to the power of 0.5.

Once these issues are solved, the paper can be published.

Comments on the Quality of English Language

The English writing could benefit from further polishing to enhance clarity and readability. 

Author Response

Dear Editor/Reviewer,

We are submitting the revised version of our manuscript titled “DocCPLNet:

Document Image Rectification via Control Point and Illumination Correction”

(Manuscript ID: [sensors-3796052]). We sincerely thank you for the valuable

feedback, which has significantly improved the quality of our work.

In response to your comments, we have made the following revisions:

  1. Notation consistency (Line 210): We have replaced the dual-index no

tation p(i, j) with the single-index notation pi , aligning with the definition in

Equation (1). The revised sentence now reads:

“Each point pi = (x, y) on the 31×31 grid corresponds to a deformation-sensitive

region, ensuring dense coverage of document boundaries and text lines.”

  1. Equation (1) corrections: We have thoroughly revised Equation (1) to

address the identified issues:

  • Added the missing absolute value bars in the condition
  • Replaced the undefined expression with the correct variable ˆpi
  • Fixed the mathematical operation in the else clause to subtraction

The corrected equation is now:

LsmoothL 1 =

1

Nc

Nc

X

i=1

( 0.5 (p

i

ˆpi

) 2

,

if |pi ˆpi | < 1

|p

i

ˆp

i

| −

0

.5

,

otherwise

(1)

We have carefully reviewed the entire manuscript to ensure these corrections

have been properly implemented and that no similar issues remain. We believe

the revised manuscript now addresses all concerns raised and meets the journal’s

standards for publication.

Thank you for the opportunity to improve our work. Please let us know if any

further revisions are required.

Sincerely,

Hongyin Ni, Jiayu Han, and Co-authors Northeast Electric Power University

Jilin, China Email: 2202301046@neepu.edu.cn
